# Data-Efficient Variational Mutual Information Estimation via Bayesian Self-Consistency

**Desi R. Ivanova**
University of Oxford, UK

**Marvin Schmitt**
University of Stuttgart, Germany

**Stefan Radev**
Rensselaer Polytechnic Institute, USA

## Abstract

Mutual information (MI) is a central quantity of interest in information theory and machine learning, but estimating it accurately and efficiently remains challenging. In this paper, we propose a novel approach that exploits Bayesian self-consistency to improve the data efficiency of variational MI estimators. Our method incorporates a principled variance penalty that encourages consistency in marginal likelihood estimates, ultimately leading to more accurate MI estimation and posterior approximation with fewer gradient steps. We demonstrate the effectiveness of our method on two tasks: (1) MI estimation for correlated Gaussian distributions; and (2) Bayesian experimental design for the Michaelis-Menten model. Our results demonstrate that our self-consistent estimator converges faster whilst producing higher quality MI and posterior estimates compared to baselines.

## 1 Introduction

Estimating dependencies between random variables is a fundamental problem in science and engineering. Mutual information (MI) is a general information-theoretic metric to quantify pairwise relationships that manifest themselves beyond the covariance matrix [15]. Additionally, estimating and optimizing MI is fundamental to many machine learning problems and tasks, such as representation learning [5, 28], understanding neural networks [26], Bayesian optimization and active learning [10, 14, 18], and Bayesian experimental design [17, 23], to name just a few.

Despite its favorable theoretical properties, estimating MI involves nesting of expectations, rendering the problem *doubly intractable* [22, 29]. As a result, its optimization is extremely challenging, especially in high-dimensional settings [27]. Recent advances in representation learning have proposed computationally tractable variational surrogate objectives, parameterized by neural networks [2, 20, 21, 28]. These approaches promise improved scalability not only in terms of dimensionality but also in handling complex data distributions and large-scale datasets. However, as McAllester & Stratos [19] show, MI estimation becomes challenging not only with increasing dimensionality, but also with the magnitude of the MI itself. They prove that any distribution-free estimator requires an exponential number of samples, which clearly does not scale in a computationally feasible way.

In many applications of probabilistic modeling, however, we have access to at least one closed-form distribution. In Bayesian analysis, this could be the prior distribution over a set of parameters $\theta$ which expresses our knowledge about plausible (latent) parameter values before observing data. In such cases, the limitations of distribution-free estimators [19, 25] no longer apply, opening up opportunities for more efficient estimators. In line with this research stream, we investigate a novel self-consistency penalty that has the potential to improve the sample efficiency and estimation quality of variational MI estimators. By encouraging *self-consistency* [16, 24] in the probabilistic joint model $p(x, y)$,

Workshop on Bayesian Decision-making and Uncertainty, 38th Conference on Neural Information Processing Systems (NeurIPS 2024).

even when parts of the model are approximated with surrogate distributions, we demonstrate that we can further enhance the sample efficiency of variational MI estimators. This penalty is designed as a simple plug-in for existing variational estimators, requiring minimal modifications to current implementations. The trade-off for this improved efficiency is a slight increase in computational cost *during training*, though importantly, it does not require additional samples from real-world observations or expensive simulators or extra computation during inference time.

## 2   Background

Mutual information (MI) is a fundamental concept in information theory, measuring the amount of information shared between two random variables. For random variables $X$ and $Y$, the MI is the Kullback-Leibler (KL) divergence between their joint probability and the product of their marginals:

$$I(X;Y) = \mathbb{E}_{p(x,y)} \left[ \log \frac{p(x,y)}{p(x)p(y)} \right] = \mathbb{E}_{p(x,y)} \left[ \log \frac{p(y \mid x)}{p(y)} \right] = \mathbb{E}_{p(x,y)} \left[ \log \frac{p(x \mid y)}{p(x)} \right]. \quad (1)$$

Estimating MI in practice is computationally costly, particularly when dealing with high-dimensional or continuous variables, with various estimation techniques developed to address these challenges.

If one of the conditional distributions, for instance, $p(y \mid x)$, is available in closed-form, we can approximate the marginal distribution $p(y) = \mathbb{E}_{p(x)}[p(y \mid x)]$ using $L$ samples $x_l \sim p(x)$. This gives us the Nested Monte Carlo (NMC) estimator (also known as double-loop Monte Carlo or DLMC):

$$\mathcal{U}^{\mathrm{NMC}}(L) := \mathbb{E}_{x_0,y_0,x_{1:L} \sim p(x_0)p(y|x_0) \prod_\ell p(x)} \left[ \log \frac{p(y_0 \mid x_0)}{\frac{1}{L} \sum_{\ell=1}^{L} p(y_0 \mid x_\ell)} \right]. \quad (2)$$

The concavity of $\log(\cdot)$ means that NMC is biased, providing an upper bound on the MI. To obtain a corresponding lower bound and thus "sandwich" the true MI, we can modify the NMC estimator by including the probability of the "positive" sample from the numerator to the "negative" (or contrastive) samples in the denominator. This leads to the Prior Contrastive Estimator (PCE [9]):

$$\mathcal{L}^{\mathrm{PCE}}(L) := \mathbb{E}_{x_0,y_0,x_{1:L} \sim p(x)p(y|x_0) \prod_\ell p(x)} \left[ \log \frac{p(y_0 \mid x_0)}{\frac{1}{L+1} \sum_{\ell=0}^{L} p(y_0 \mid x_\ell)} \right]. \quad (3)$$

Both bounds are asymptotically consistent, converging to the the true MI in the limit as $L \to \infty$. Optimal convergence efficiency is achieved by setting $L \propto \sqrt{N}$, where $N$ is the number of samples we use to approximate the outer expectation. This, at best, gives us an overall convergence rate of $\mathcal{O}(C^{-1/3})$, where $C = NL$ is the total computational budget, which is considerably worse than $\mathcal{O}(C^{-1/2})$ for conventional Monte Carlo. One way to improve the efficiency of these basic estimators is to introduce a variational proposal distribution $q_v(x \mid y)$ and apply importance sampling to the $x$ values, potentially leading to a more accurate approximation of the marginal $p(y)$:

$$p(y) \approx \frac{1}{L} \sum_{\ell=1}^{L} \frac{p(x_\ell)p(y \mid x_\ell)}{q_v(x_\ell \mid y)} \quad (4)$$

Substituting Eq. 4 into the denominator of Eq. 2 yields the Variational Nested Monte Carlo (VNMC) estimator, which we denote by $\mathcal{U}^{\mathrm{VNMC}}(q_v; L)$. Similarly, adjusting for the positive sample and substituting in Eq. 3, we obtain the Variational Prior Contrastive Estimator (VPCE), which we denote as $\mathcal{L}^{\mathrm{VPCE}}(q_v; L)$. For clarity, the estimators are given in Appendix A.

These variational estimators preserve the upper and lower bound properties of their non-variational counterparts whilst offering an improved estimation efficiency. Specifically, in addition to asymptotic consistency as $L \to \infty$, VNMC and VPCE can also achieve finite sample unbiasedness (with $L = 1$) for the *optimal* proposal, which equals the posterior $q_v^*(x_\ell \mid y) = p(x_\ell \mid y)$.

## 3   Method

By rearranging Bayes' rule, we can express the marginal distribution (i.e., the *evidence* in Bayesian inference) of $y$ as $p(y) = p(x) \, p(y \mid x) \, p(x \mid y)^{-1}$. An important observation that follows directly is the *self-consistency* of the marginal $p(y)$ when evaluated based on different values of $x$. In other words, for any set of values $x_1, \dots x_K$ we have:

$$\frac{p(x) \, p(y \mid x)}{p(x \mid y)} = \mathrm{const} \, \forall x \quad \implies \quad \frac{p(x_1) \, p(y \mid x_1)}{p(x_1 \mid y)} = \dots = \frac{p(x_K) \, p(y \mid x_K)}{p(x_K \mid y)} \quad (5)$$

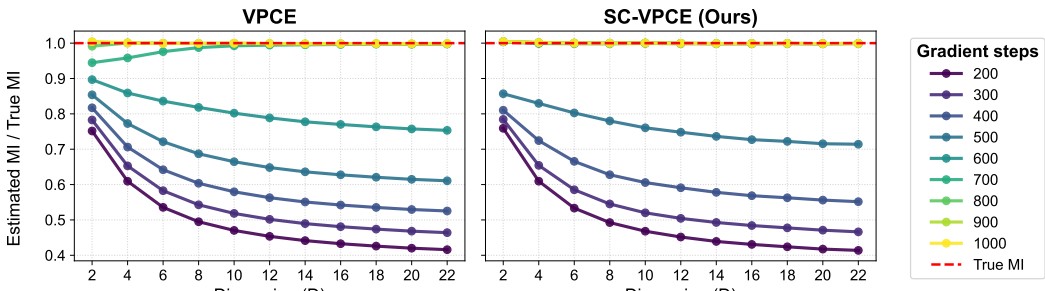

Figure 1: **Correlated Gaussian Distributions.** Comparison of MI estimation accuracy between baseline VPCE and our self-consistent VPCE (SC-VPCE) across various dimensions. Our method achieves near-perfect MI estimation with approximately $25\%$ fewer gradient steps. Results shown use a batch size of $N = 16$, $L = 4$ negative samples, and $K = 8$ self-consistency samples during training. Each curve represents the average of five independent runs.

However, if we replace the posterior $p(x \mid y)$ with an approximation $q(x \mid y)$, Eq. 5 will not hold exactly, with different $x_k$ yielding different $p(y)$ values. This variance is undesirable, as it is a direct result of *approximation error*. Thus, a high-quality approximate posterior $q(x \mid y)$ minimizes the variance of $p(y)$. Recently, [24] used this insight to propose a variance penalty that improves the sample efficiency of neural posterior estimation in amortized Bayesian inference,

$$\mathcal{L}_{\text{SC}}(q_v; y) := \text{Var}_{\tilde{x} \sim q(x)}\Big( \log p(\tilde{x}) + \log p(y \mid \tilde{x}) - \log q_v(\tilde{x} \mid y) \Big), \qquad (6)$$

where $q(x)$ is an easy-to-sample proposal distribution whose choice has non-trivial consequences for the empirical behavior of Eq. 6. Building on these results, we explore the utility of self-consistency in the context of MI estimation and optimization.

## 4 Empirical Evaluation

In this section, we report preliminary results validating our self-consistent estimation algorithm. Our experiments focus on two key areas: mutual information estimation and Bayesian optimal experimental design. The main baseline we compare against is the VPCE estimator. We use identical neural network architectures and hyperparameters for all methods to ensure a fair comparison. To provide a controlled setting for these initial investigations, we assume access to an analytic likelihood function throughout our experiments. We note that a likelihood function can be learned within the same optimization scheme [24], and we plan to extend our approach to such scenarios in future work.

**Metrics** The main focus on two key aspects of MI estimators: accuracy and data efficiency, particularly in high MI regimes. Whilst quick and accurate MI estimation is desirable in the context of optimization, obtaining high-quality amortized posteriors is also of paramount interest due to their practical utility in downstream tasks, such as marginal likelihood estimation or cross-validation. To this end, we use the squared maximum mean discrepancy (MMD; [11]) distance between a "ground-truth" reference posterior $p(x \mid y)$, estimated with HMC [3, 12], and our approximation $q(x \mid y)$:

$$\text{MMD}^2(p \,||\, q) = \mathbb{E}_{x, x' \sim p(x|y)}[\kappa(x, x')] + \mathbb{E}_{x, x' \sim q(x|y)}[\kappa(x, x')] - 2\mathbb{E}_{x \sim p(x|y), x' \sim q(x|y)}[\kappa(x, x')], \ (7)$$

where $\kappa(\cdot, \cdot)$ is a positive definite kernel. Further evaluation metrics encompass (1) the relative error in estimated mutual information; and (2) the estimation error of the log posterior density $p(x \mid y)$.

### 4.1 Mutual Information Estimation: Correlated Gaussian Distributions

This experiment serves as a proof-of-concept in a controlled setting where closed-form values for all quantities of interest are known, allowing for a principled evaluation. We follow a similar experimental setup to that in Poole et al. [21] and consider $D$-dimensional correlated Gaussian random vectors $X$ and $Y$ with some correlation $\rho$.

**Setup** We evaluate both VPCE and our self-consistent VPCE across various dimensionalities $D \in \{2, 4, \ldots, 20\}$ and $\rho = 0.98$, resulting in MIs between 3.0 and 36. We train the estimators with batch size of 16 and set the number of negative samples to $L = 4$. We use $K = 8$ self-consistency samples to estimate the variance in Eq. 6. We track the estimated mutual information

and posterior quality over the course of 1000 gradient steps. We train a simple Gaussian proposal with a learnt mean and standard deviation: $q_v(\cdot \mid y) = \mathcal{N}(\cdot \mid \alpha + \beta\, y, \sigma)$. To ensure robustness, we repeat the training and evaluation procedures five times with different seeds and report the averages.

**Results** As shown in Figure 1, the VPCE baseline requires about 800 gradient steps to achieve accurate MI estimates. In contrast, our SC-VPCE achieves the same performance with only 600 gradient steps, which amounts to a $25\%$ relative increase in sampling efficiency. Furthermore, since we have access to the true posterior in closed form for this analytical experiment, we can compute the KL divergence exactly, providing a precise measure of posterior quality. Figure 2 illustrates that our self-consistent VPCE yields essentially perfect approximations of the posterior distribution (measured by KL divergence) while being approximately $50\%$ more sample-efficient than the VPCE baseline.

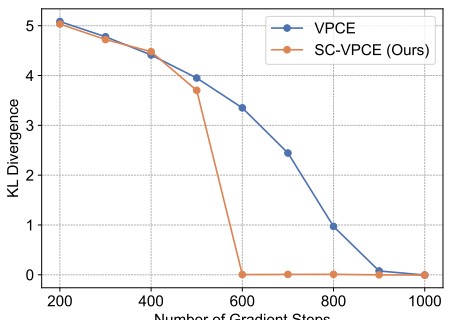

Figure 2: **Correlated Gaussians.** Our SC-VPCE converges much faster to the true posterior, as measured by KL divergence.

## 4.2 Bayesian Experimental Design: Michaelis-Menten Model

Originally rooted in statistics [4, 17], Bayesian experimental design (BED), has recently seen renewed interest, particularly in the engineering and machine learning communities [1, 8, 13, 23]. The objective in BED is to choose experiments that yield the most informative data about underlying model parameters $\theta$. Denoting the experiment parameters by $\xi$ and the outcome of the experiment by $y$, the optimal design $\xi^*$ maximizes the mutual information between $y$ and $\theta$, i.e. $\xi^* = \arg\max_\xi I(\theta; y \mid \xi)$.

We consider a static experimental design problem, where the goal is to learn the parameters in the Michaelis-Menten model [6]. Michaelis-Menten is a non-linear model that is widely used in biology and chemistry to describe the relationship between the (scaled) concentration of a substrate $\xi \in [0, 1]$ and the rate of an enzymatic reaction $\theta = (\theta_1, \theta_2) \in \mathbb{R}^2$. The outcome $y$ of an experiment with design $\xi$ is modelled probabilistically as $y \mid \xi, \theta \sim \mathcal{N}(f(\xi; \theta), \sigma^2)$ with $f(\xi; \theta) = \theta_1 (\xi S)^b / \theta_2^b + (\xi S)^b$, where $S$ and $b$ are a fixed scaling factors. The prior on the parameters of interest is $p(\theta_1) = \mathcal{N}(\theta_1; 0.5, 0.1^2)$, $p(\theta_2) = \mathcal{N}(\theta_2; 0.5, 0.1^2)$.

**Setup** We use the SC-VPCE lower bound to learn 10 designs $\xi_1, \ldots, \xi_{10}$ using a batch size of $N = 32$, $L = 32$ contrastive samples, and vary the number of self-consistency samples $K \in [64, 32, 16, 0]$, with $K = 0$ corresponding to the VPCE baseline. We use $b = 6$, $S = 400$, and observation noise $\sigma = 10.0$. We choose an expressive neural-based proposal distribution, namely a neural spline flow (NSF) [7] that learns an invertible transformation from a Gaussian base distribution to the target posterior distribution $p(x \mid y)$. The posterior density is tractable via the change-of-variables formula [16]. The NSF $q_\nu$ is parameterized by a total of $5\,980$ learnable parameters $\nu$, spread across 2 neural transformation blocks with 32 hidden features. MMD computations are performed over 50 simulations, using the learned design.

| Gradient steps | $K$ | MMD ($\times 10^{-3}$) | MI |
|---|---|---|---|
| 500 | 64 | 7.28 | 2.721 |
| 500 | 32 | 7.49 | 2.723 |
| 500 | 16 | 7.24 | 2.729 |
| 500 | 0 | 10.23 | 2.766 |
| 1000 | 64 | 4.14 | 2.764 |
| 1000 | 32 | 4.06 | 2.724 |
| 1000 | 16 | 5.24 | 2.753 |
| 1000 | 0 | 6.49 | 2.836 |
| 2000 | 64 | 3.98 | 2.750 |
| 2000 | 32 | 2.55 | 2.723 |
| 2000 | 16 | 3.42 | 2.726 |
| 2000 | 0 | 4.28 | 2.837 |

Table 1: **Michaelis-Menten Model.**

**Results** As the number of gradient steps increases from 500 to 2000, we observe a general trend of decreasing MMD values, indicating improved posterior quality (see Table 1). Our method consistently achieves lower MMD values, indicating superior posterior estimates. We observe that the MI values remain relatively stable, suggesting that the designs quickly converge to their optimum.

## 5 Conclusion

In this paper, we presented a method to integrate self-consistency losses into variational MI estimators. Compared to baselines, our data-efficient estimator requires fewer gradient steps to converge and

produces better posterior distributions, which can be useful for downstream tasks. In future work we plan to reduce the required modeling assumptions by extending our method to learned likelihoods.

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

## A  VNMC and VPCE

The variational nested Monte Carlo (VNMC) upper bound is given by:

$$\mathcal{U}^{\text{VNMC}}(q_v; L) := \mathbb{E}_{x_0, y_0, x_{1:L} \sim p(x)p(y|x_0) \prod q_v(x_{1:L}|y_0)} \left[ \log \frac{p(y_0 \mid x_0)}{\frac{1}{L} \sum_{\ell=1}^{L} \frac{p(x_\ell)p(y_0|x_\ell)}{q_v(x_\ell|y_0)}} \right]. \qquad (8)$$

The corresponding variational PCE lower bound is given by:

$$\mathcal{L}^{\text{VPCE}}(q_v; L) := \mathbb{E}_{x_0, y_0, x_{1:L} \sim p(x)p(y|x_0) \prod q_v(x_{1:L}|y_0)} \left[ \log \frac{p(y_0 \mid x_0)}{\frac{1}{L+1} \sum_{\ell=0}^{L} \frac{p(x_\ell)p(y_0|x_\ell)}{q_v(x_\ell|y_0)}} \right]. \qquad (9)$$

## B  Further Experiment Details Results

### B.1  Mutual Information Estimation: Correlated Gaussian Distributions

**Proposal parameterisation**   Since the true posterior is Gaussian, we train a simple Gaussian proposal distribution with a learnt mean and standard deviation, i.e. $q_v(\cdot \mid y) = \mathcal{N}(\cdot \mid \alpha + \beta y, \sigma)$, where $\alpha, \beta$ and $\sigma$ are learnt parameters.

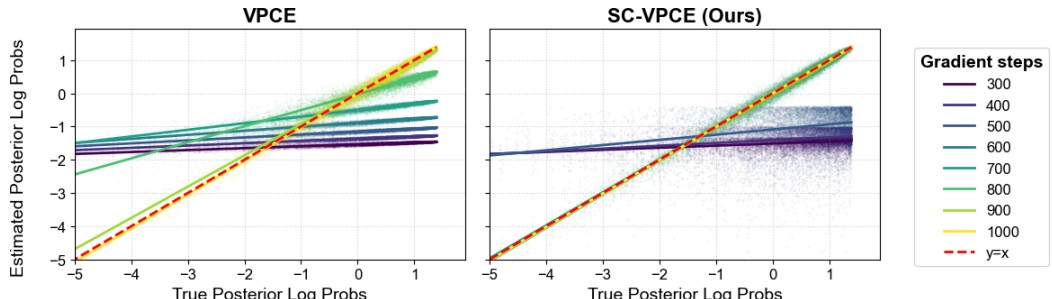

Figure 3: **Correlated Gaussians**: Estimated, $q_v(x \mid y)$, and actual, $p(x \mid y)$, posterior log-probabilities computed on a test set.

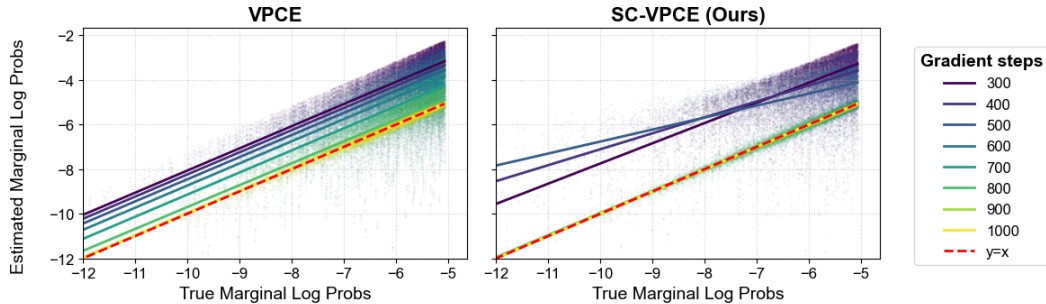

Figure 4: **Correlated Gaussians.** Estimated, $\log \hat{p}(y) = \log p(x) + \log p(y \mid x) - \log q_v(x \mid y)$ and actual, $p(y)$, marginal log-probabilities computed on a test set.

