# OpenReview forum: "Data-Efficient Variational Mutual Information Estimation via Bayesian Self-Consistency"
_NeurIPS.cc/2024/Workshop/BDU — NeurIPS BDU Workshop 2024 Poster_

### Official Review · Reviewer_2tGq · 2024-09-24
**Review for "Data-Efficient Variational Mutual Information Estimation via Bayesian Self-Consistency"**

**Rating:** 6
**Confidence:** 4

**Review:**

The paper proposes a bound for use in mutual information maximization that can be an alternative to VNMC and VPCE.


Major comments:
- The self consistency explanation for the derivation of the bound (although the basic idea is proposed in a cited paper and not here, I am going to criticise it here) is informal, perhaps too informal (admittedly a subjective value judgement). For example, it leads to statements like that in line 76-77: " Thus, a high-quality approximate posterior q(x | y) minimizes the variance of p(y). " p(y) is either a number (scalar) or a PDF, function of y, not a random variable, so it does not have a variance. The way I can make sense of this self consistency is that it is a more vague way of saying that we would like low-variance estimates of p(y) like with the estimator in Eq. (4), i.e. an importance sampling estimator. And, well, that we already know from standard MC theory - yes, in Eq. (4) we want low variance, amounting to the q being close to the true posterior ( and variance = chi^2 divergence ). Therefore the jump to Eq. (6) is too loosely motivated.
- It should also be mentioned that the proposed loss of Eq (6) appears before for instance in Richter, L. et al (2020) (see Eq. (5)), although in a different context (gradient estimation).
- The full paper should say how you plan to maximize and / or estimate Eq 6, conditions under which your (gradient?) estimators have finite variance, and how this compares with VNMC/VPCE.
- An interesting question is that now you have two distributions to choose/optimize: q(x) and q_v(x | y), while e.g. VNMC/VPCE only have the latter. How can we compare apples to apples here? You should elaborate on this. What should q(x) be? if you do a simple MC estimate of Eq (6) sampling from q(x) , by IS theory (see e.g. Owen's "Monte Carlo theory, methods and examples" (2013) ) the optimal q(x) to estimate Eq (6) is a weird function. Or maybe you want a q(x) that minimizes the variance of a *gradient* estimator of Eq (6). All things that should be elaborated upon. And how to intertwine with the optimization of q_v(x|y) which supposedly should just try to get close to p(x|y).
- In the metrics, "the estimation error of the log posterior density" - what do you mean precisely ? I am guessing you mean for some samples / values x_1, etc. you calculate  approximation(x1) vs true_density(x1) and see how far they are. This is a weird metric, if so. Better to have things like estimates of divergences. Also the MMD is a bit arbitrary / fishy since it depends on an arbitrary kernel.  Error in mutual info estimation makes more sense, I would suggest plotting boxplots of estimates vs true value (if not closed form, obtained with a very long run of a consistent method).  Although also not a fully relevant metric if the aim is to *maximize* MI as opposed to *estimate it*, in which case some notion of convergence speed to the minimum should be there.

I have to say that I personally do not see what the "self consistency" perspective brings to arriving to Eq 6, which maybe could have a more formal motivation (it could borrow from Richter, L. et al 2020 but not necessarily).

Refs.

- Richter, L., Boustati, A., Nüsken, N., Ruiz, F. and Akyildiz, O.D., 2020. VarGrad: a low-variance gradient estimator for variational inference. Advances in Neural Information Processing Systems, 33, pp.13481-13492.

---

### Official Review · Reviewer_38ok · 2024-10-07
**Evaluation of Data-Efficient Variational Mutual Information Estimation via Bayesian Self-Consistency: Insights on Methodology and Application**

**Rating:** 8
**Confidence:** 5

**Review:**

The manuscript introduces a novel approach to estimating mutual information (MI) by harnessing Bayesian self-consistency, representing a considerable advancement in the landscape of information theory and machine learning. The authors rigorously derive their variational estimator, incorporating a principled variance penalty that encourages consistency in marginal likelihood estimates. This technical innovation effectively addresses the challenges associated with MI estimation, particularly in high-dimensional spaces where conventional methods exhibit significant sample inefficiency. The rigorous empirical evaluation, demonstrated through tasks such as MI estimation for correlated Gaussian distributions and Bayesian experimental design with the Michaelis-Menten model, provides compelling evidence that the proposed self-consistent estimator converges faster and yields more accurate MI and posterior approximations compared to existing baselines.

In terms of clarity, the manuscript is generally well-structured, facilitating a logical progression from foundational concepts to detailed methodological exposition. However, the complexity of the mathematical derivations and algorithms, particularly those surrounding Nested Monte Carlo (NMC) estimators and Variational Prior Contrastive Estimators (VPCE), can pose challenges for readers without a strong foundation in probabilistic modeling and Bayesian inference. Increased elaboration on crucial equations and the underlying assumptions would enhance understanding and accessibility. Furthermore, a more thorough explanation of experimental settings and performance metrics—such as the squared maximum mean discrepancy (MMD) and KL divergence—would provide deeper insights into the validity of the results.

The originality of this research is evident in its innovative application of self-consistency penalties to variational MI estimators, which not only enhances sample efficiency but also maintains the desirable properties of asymptotic consistency. The empirical results underline the practical significance of the findings, especially given the increasing dimensionality of datasets common in machine learning tasks. While the focus on two specific experimental setups demonstrates the method's efficacy, expanding the empirical evaluation to include a wider variety of applications would strengthen the claims regarding generalizability. Overall, this manuscript demonstrates substantial technical merit and presents a strong contribution to the field, warranting recognition for its innovative approach and comprehensive validation of results.

---

### Decision · Program_Chairs · 2024-10-09

**Decision:**

Accept (Poster)

**Comment:**

Both reviews are positive, though the first is suspiciously so and includes language that is otherwise atypical in a review (“innovative”, “underline the practical significance”, ..). Nonetheless, the second review is also positive, so I recommend acceptance.